# Antibacterial Effect of Cell-Free Supernatant from *Lactobacillus pentosus* L-36 against *Staphylococcus aureus* from Bovine Mastitis

**DOI:** 10.3390/molecules27217627

**Published:** 2022-11-07

**Authors:** Gengchen Wang, Hong Zeng

**Affiliations:** 1School of Basic Medicine, Youjiang Medical University for Nationalities, Baise 533000, China; 2Key Laboratory of Protection and Utilization of Biological Resources in Tarim Basin of Xinjiang Production & Construction Corps, College of Life Science and Technology, Tarim University, Alar 843301, China

**Keywords:** antibacterial activity, anti-bacterial mechanism, *Lactobacillus pentosus*, *Staphylococcus aureus*, biofilm

## Abstract

This study sought to analyze the main antibacterial active components of *Lactobacillus pentosus* (*L. pentosus*) L-36 cell-free culture supernatants (CFCS) in inhibiting the growth of *Staphylococcus aureus* (*S. aureus*), to explore its physicochemical properties and anti-bacterial mechanism. Firstly, the main antibacterial active substance in L-36 CFCS was peptides, which inferred by adjusting pH and enzyme treatment methods. Secondly, the physicochemical properties of the antibacterial active substances in L-36 CFCS were studied from heat, pH, and metal ions, respectively. It demonstrated good antibacterial activity when heated at 65 °C, 85 °C and 100 °C for 10 and 30 min, indicating that it had strong thermal stability. L-36 CFCS had antibacterial activity when the pH value was 2–6, and the antibacterial active substances became stable with the decrease in pH value. After 10 kinds of metal ions were treated, the antibacterial activity did not change significantly, indicating that it was insensitive to metal ions. Finally, scanning electron microscopy, transmission electron microscopy and fluorescence probe were used to reveal the antibacterial mechanism of *S. aureus* from the aspects of cell morphology and subcellular structure. The results demonstrated that L-36 CFCS could form 1.4–2.3 nm pores in the cell membrane of *S. aureus*, which increased the permeability of the bacterial cell membrane, resulting in the depolarization of cell membrane potential and leakage of nucleic acid protein and other cell contents. Meanwhile, a large number of ROS are produced and accumulated in the cells, causing damage to DNA, and with the increase in L-36 CFCS concentration, the effect is enhanced, and finally leads to the death of *S. aureus*. Our study suggests that the main antibacterial active substances of L-36 CFCS are peptides. L-36 CFCS are thermostable, active under acidic conditions, insensitive to metal ions, and exhibit antibacterial effects by damaging cell membranes, DNA and increasing ROS. Using lactic acid bacteria to inhibit *S. aureus* provides a theoretical basis for the discovery of new antibacterial substances, and will have great significance in the development of antibiotic substitutes, reducing bacterial resistance and ensuring animal food safety.

## 1. Introduction

Bovine mastitis is a very common disease that has a serious impact on animal health and economy, which causes huge production loss to the dairy industry all over the world every year [1,2]. *Staphylococcus aureus* (*S. aureus*) is one of the main pathogens causing bovine mastitis. The cure rate of bovine mastitis caused by *S. aureus* is low, and it is easy to relapse, and even make cows lose the function of lactation [3]. Antibiotics are the main means to treat cow mastitis. With the abuse of antibiotics, the pathogenic bacteria demonstrated different degrees of drug resistance [4], such as methicillin resistance *S. aureus* (MRSA). Therefore, it is of great significance to explore antibiotic substitutes, treat bacterial infections by biological control and reduce bacterial resistance to ensure the healthy development of the dairy farming industry and ensure the safety of animal food.

Lactic acid bacteria (LAB) is generally regarded as a kind of safe microorganism and has long been used in the fields of food processing [5] animal feed [6] and medical health [7,8]. LAB produces organic acids, bacteriocin, hydrogen peroxide and other metabolites with natural antibacterial activity. The researchers found that LAB inhibited the growth of MRSA [9,10], *Pseudomonas aeruginosa* (PA) [11], and Vancomycin-resistant *enterococcus* (VRE) by producing antimicrobial substances [11]. Some LAB play a major antibacterial role by producing organic acids, such as *L. plantarum* DY-6 [12], while others are able to produce H_2_O_2_ [13] to inhibit bacterial growth. In recent years, more and more researchers have discovered bacteriocins from the products of *Lactobacillus* and studied their antibacterial mechanism. Pentocin ZFM94 [14] has broad-spectrum antibacterial activity and is produced by *L. pentosus* ZFM94, which kills bacteria by disrupting cell membrane and leaking cellular content. Pentocin MQ1 [15] is a novel bacteriocin produced by *L. pentosus* CS2, which has broad-spectrum antibacterial activity and kills target bacteria by forming holes in the cell membrane. Pentocin JL-1 [16], produced by *L. pentosus* JL-1, is a new bacteriocin with broad-spectrum antibacterial activity. It has a significant effect on MRSA, and its action site is the cell membrane. All of these bacteriocins are sensitive to proteases.

In this study, *L. pentosus* L-36 demonstrated strong antibacterial activity. We investigated the main antibacterial components in CFCS and their stability to pH, temperature and metal ions. By means of scanning electron microscopy, transmission electron microscopy and fluorescence probe, the antibacterial mechanism of L-36 CFCS was revealed from the aspects of cell morphology and the subcellular structure.

## 2. Results

### 2.1. Antibacterial Activity against Clinical Isolates of S. aureus

The antibacterial activities of L-36 CFCS against standard strain and 10 clinical isolates (Table 1). L-36 CFCS displayed similar strong antibacterial activity toward standard strains and 10 clinical isolates.

### 2.2. Effect of CFCS on the Dynamic Growth of the S. aureus

The lag phase of *S. aureus* was extended to 10 h when L-36 CFCS with the 1/2 × MIC concentration were treated. However, the growth of *S. aureus* had little effect when L-36 CFCS with the 1/4 × MIC and 1/8 × MIC concentration were treated (Figure 1).

### 2.3. Analysis of Antibacterial Active Substances in CFCS

The antibacterial active substances produced by LAB with lactic acid, which was the same as the initial CFCS pH, had no antibacterial activity. At the same time, when the pH of CFCS was 5.5, 70.14% of the antibacterial activity was still found, indicating that the antibacterial active substances were not organic acids. Then, the catalase and peroxidase were used to treat CFCS to determine whether the antibacterial active substances were H_2_O_2_. The results demonstrated that CFCS were sensitive to catalase and peroxidase, indicating that H_2_O_2_ was one of the antibacterial active substances. Finally, CFCS were treated with protease and peptidase to determine whether the antibacterial active substances were peptide; the result turns out that CFCS were sensitive to all proteases except pepsin, of which trypsin was the most sensitive. Bacteriocins produced by lactic acid bacteria are sensitive to protease, especially to trypsin [17]; thus, it is speculated that the peptide produced by L-36 might be bacteriocins and might play a major antibacterial role (Figure 2).

### 2.4. Effects of Heat, pH and Metal Ions on Antibacterial Activity

The effects of temperature, pH and metal ion on the antibacterial activity of L-36 CFCS were determined (Figure 3). The antibacterial activity of L-36 CFCS was not significantly different from that of the control group—either the heating temperature increased or the time was prolonged (Figure 3a). The results demonstrated that L-36 CFCS was strongly thermostable. The initial pH of the L-36 CFCS is about 3.5. When the pH of L-36 CFCS was adjusted to 2 (less than the initial L-36 CFCS), the antibacterial activity was significantly increased. However, the antibacterial activity decreased successively when the pH of L-36 CFCS was adjusted to 4–6 (greater than the initial L-36 CFCS). When the pH values of L-36 CFCS were adjusted to 7 and 8 (greater than the initial L-36 CFCS), the antibacterial activity disappeared.

The antibacterial effect of pH 2 and 3 MRS broth medium (adjust with lactic acid) was significantly weaker than that of L-36 CFCS, and disappeared when the pH value was greater than 3. Results indicated that the antibacterial active substances in L-36 CFCS have antibacterial activity between pH 2 and 6. With the decrease in pH value, the antibacterial active substances become more stable and their antibacterial activity increases (Figure 3b). Different types and different concentrations of metal ions had no effects on the antibacterial ability of L-36 CFCS (Figure 3c).

### 2.5. Mode of Action of L-36 CFCS

#### 2.5.1. SEM and TEM

Microstructure and intracellular structure of *S. aureus* treated with 1 × MIC L-36 CFCS were observed by SEM and TEM. According to SEM, the normal growth of *S. aureus* demonstrated smooth and intact surface with regular morphology and uniform size (Figure 4a). When incubated with L-36 CFCS, the membrane surfaces of *S. aureus* became shrunken, bumpy and damaged, which caused the intracellular substances to leak and die (Figure 4b,c). The effect of L-36 CFCS on *S. aureus* cells was also observed using TEM. Control group cell exhibited normal morphology, a complete cell wall and a cell membrane (Figure 4d). Meanwhile, the cell membrane obviously shrunk inward and was broken when treated with L-36 CFCS (Figure 4e,f).

#### 2.5.2. Effect of CFCS on Cell Membrane Integrity

Both fluorescence microscopy and automatic microplate reader were used to investigate the membrane integrity of *S. aureus* after treatment with different concentrations of L-36 CFCS using PI as the fluorescent marker (Figure 5A). The fluorescence microscopy results demonstrated that the untreated *S. aureus* emitted nearly no fluorescence. As the concentration of L-36 CFCS increased, *S. aureus* emitted more red fluorescence. The fluorescence intensity was tested by automatic microplate reader, and the fluorescence intensity was detected in *S. aureus* cells with different concentrations of L-36 CFCS, which were significantly higher than untreated *S. aureus* cells, especially the 4 × MIC group (Figure 5A(a5)).

#### 2.5.3. Estimation of Pore Size in Cytomembrane of *S. aureus*

The pore size of the cell membrane of *S. aureus* treated with L-36 CFCFS was measured by using FD4 and FD10 as the fluorescent marker (Figure 5B). Fluorescence microscopy results demonstrated that no green fluorescence was observed in *S. aureus* that were untreated with L-36 CFCFS, but it was observed in *S. aureus* treated with 1 × MIC concentration of L-36 CFCFS. Fluorescence intensity were tested by the automatic microplate reader; the results demonstrated that the fluorescence intensity was detected in *S. aureus* cells with 1 × MIC concentration of L-36 CFCS, which were significantly higher than untreated *S. aureus* cells (Figure 5B(b5)).

#### 2.5.4. Effects of CFCS on Leakage of Nucleic Acid and Proteins in *S. aureus*

The results revealed that L-36 CFCS caused the leakage of nucleic acids and proteins (Figure 5C) from treated *S. aureus*, and the increase in protein was very similar to the increase in nucleic acids. 

OD_260_ and OD_280_ of *S. aureus* treated with 2 × MIC L-36 CFCS increased to maximum at 12 h, followed by a steady state. When *S. aureus* treated with 4 × MIC L-36 CFCS, OD_260_ and OD_280_ increased to maximum at 8 h, followed by a decline state. Additionally, no obvious changes in the OD_260_ and OD_280_ of the 1 × MIC group and the control group were observed.

#### 2.5.5. Effects of CFCS on Membrane Potential of *S. aureus*

To explore whether the L-36 CFCS affected the membrane potential of the *S. aureus*, DiSC_3_(5), as the fluorescent marker, was used in the membrane depolarization experiments. According to the automatic microplate reader results, different concentrations of L-36 CFCS significantly increased the fluorescence intensity of *S. aureus* (Figure 5D).

#### 2.5.6. Effects on Subcellular Structure of *S. aureus* by L-36 CFCS

DAPI was used as a fluorescent marker to measure the intracellular DNA damage of *S. aureus* treated with different concentrations of L-36 CFCS (Figure 6A). Fluorescence microscopy results demonstrated that blue fluorescence appeared in the untreated group, while brilliant blue fluorescence appeared when treated with different concentrations of L-36 CFCS (circled in the Figure 6A), indicating that the intracellular DNA was damaged, and the number of damaged cells increased with the increase in concentration. The automatic microplate reader results demonstrated that the fluorescence intensity of the treated groups was gradually increased and significantly higher than that of the untreated group, indicating that L-36 CFCS caused DNA damage in *S. aureus* in a concentration-dependent manner (Figure 6A(a5)).

ROS production in *S. aureus* cells treated with L-36 CFCS was detected by using DCFH-DA as the fluorescent marker (Figure 6B). Fluorescence microscopy demonstrated that almost no fluorescence appeared in the untreated group, while a large number of green fluorescence appeared in the treated groups, indicating that L-36 CFCS with different concentrations induced ROS production in *S. aureus*. The automatic microplate reader results demonstrated that a small amount of fluorescence appeared in the untreated group, and the fluorescence intensity was significantly enhanced when *S. aureus* was treated with different concentrations, especially 4 × MIC (Figure 6B(b5)). It indicates that L-36 CFCS can cause ROS production in *S. aureus* cells, which was concentration dependent.

## 3. Discussion

In this study, *L. pentosus* L-36 had similar strong antibacterial effect to the clinical isolate and the standard strain. LAB exert antibacterial effects by producing organic acids, H_2_O_2_, and bacteriocins. By verifying one by one, it was found that the antibacterial active substances of L-36 were peptides and H_2_O_2_, among which peptides were the main ones.

The pH test demonstrated that L-36 CFCS was active over a range of pH (2.0–6.0), which is conducive to the stability of peptides and H_2_O_2_, making them higher antimicrobial active. The reason might be: (1) peptides had synergistic effects with acids [18]; (2) peptides had a positive charge, were more stable in an acidic environment, and easily attach to the *S. aureus* membrane [19]; H_2_O_2_ is an extremely weak acid ($H_{2}O_{2}⇌H^{+}+{\mathrm{HO}}_{2}^{-}$, *K_a_* = 2.4 × 10^−12^), the decrease in pH value will facilitate the stabilization of hydrogen peroxide, thus enhancing its antibacterial activity. 

Most bacteriocins have good thermal stability. The antibacterial activity of Bacteriocin DY4-2 remained above 98.7% after being treated at 121 °C for 30 min [20]. The activity of bacteriocin produced by *L. pentosus* MS031 could still reach more than 97% after being treated at 121 °C for 20 min [21]. The antibacterial effect of L-36 CFCS treated with 65 °C, 85 °C and 100 °C for 10 and 30 min was not significantly different from that of the control group, indicating that L-36 CFCS was thermo stable, which is the dominant characteristic of most bacteriocin. This characteristic of bacteriocins makes them widely applicable in the hot processed food industry.

The increase in metal ions may cause the disorder of the cell membrane potential and increase in the permeability of *S. aureus*, which is conducive to the antibacterial effect [22]. Meanwhile, metal ions may also bind to the active site of antibacterial substances, resulting in conformational changes and inhibiting antibacterial activity [22]. The antibacterial effect of L-36 CFCS treated with 10 metal ions was not significantly different from that of the control group, indicating that L-36 CFCS was insensitive to metal ion. It was possible that the promoting and inhibiting effects of metal ions on L-36 CFCS were neutralized, resulting in no significant change to the antibacterial effect.

The antibacterial substances produced by L-36 form pores of about 1.4–2.3 nm on the surface of bacterial cell membrane, resulting in increased permeability, leakage of intracellular nucleic acid and protein, depolarization of membrane potential, and massive production and accumulation of ROS, resulting in DNA damage in bacteria, and ultimately bacterial death. The damage of peptides to the cell membrane of the target bacteria may be caused by electrostatic interaction between the positively charged peptides and the negatively charged target bacteria cell surface, and the hydrophobic surface faces the cell membrane and passes through the lipid bilayer. After penetrating the lipid bilayer, the peptides affect gene expression and protein synthesis, leading to cell death [23]. The increased permeability of the cell membrane of target bacteria caused by peptides may also result in the dissipation of the proton dynamic potential and consumption of intracellular ATP and leakage of intracellular substances through the formation of pores, resulting in bacterial death [24].

## 4. Materials and Methods

### 4.1. Bacterial Strains and Culture Conditions 

*S. aureus* ATCC 29213 was provided by Key Laboratory of Tarim Animal Husbandry Science and Technology, Xinjiang Production & Construction Corps, Tarim University. *S. aureus* isolates from clinical bovine mastitis was isolated from a dairy farm in Southern Xinjiang, and kept by the Engineering Laboratory for Tarim Animal Diseases Diagnosis and Control of Xinjiang Production & Construction Corps. Lactic acid bacteria was offered by professor Xiao-pu ren in Tarim University. LAB and *S. aureus* were grown on MRS agar plates and TSA plates, respectively. It was stored in medium supplemented with 20% glycerol at −20 °C. Screening of antibacterial LAB was completed using an agar spot test in our previous study, in which *S. aureus* was used as indicators. Strains with strong antibacterial activity were further identified in our previous study by 16S rDNA sequencing after PCR amplification. 

### 4.2. Antibacterial Activity

L-36 was incubated in 10 mL MRS broth at 37 °C for 24 h, then transferred to 100 mL of MRS broth and incubated for 72 h under the same conditions. Subsequently, the culture was centrifuged at 13,000× *g* for 10 min at 4 °C. The supernatants were concentrated by freeze-drying and dissolved with ddH_2_O, then filtrated using 0.22 μm sterile filters to achieve CFCS.

### 4.3. Minimal Inhibitory Concentration (MIC)

The MIC of CFCS against *S. aureus* ATCC 29213 was determined by broth microdilution assay [15]. The final concentrations of CFCS ranged from 250 mg/mL to 0.489 mg/mL, and were at 37 °C for 24 h. Each concentration was conducted in triplicate. The MIC represents the CFCS concentration at which 100% of the *S. aureus* growth is inhibited. 

### 4.4. Effect of CFCS on the Dynamic Growth of the S. aureus

The *S. aureus* were inoculated in TSB at 37 °C for 24 h, and the bacteria cell was resuspended in TSB broth. Then, the CFCS was mixed with the bacterial suspension at a final concentration of 1/2 × MIC, 1/4 × MIC and 1/8 × MIC. The real-time microbial growth analysis system was used to record the dynamic growth of *S. aureus*.

### 4.5. Analysis of Antibacterial Active Components of CFCS

The antibacterial activity was detected by the double agar diffusion method. In order to analyze whether organic acids exist in CFCS to exert antibacterial activity, the pH of CFCS was adjusted to 5.5, and lactic acid was used to adjust the pH of MRS medium to the same pH as the initial CFCS, as the control. To analyze whether H_2_O_2_ and protein substances exist in CFCS to play antibacterial activity, the pH value of CFCS was adjusted to the optimum pH of catalase, peroxidase, protein K, trypsin, pepsin, papain and peptidase, respectively. Then, adding enzymes, to make the final concentration to be 1 mg/mL. After being reacted at 37 °C for 2.5 h, the pH value was adjusted to the initial value, and the initial CFCS as the control [20].

### 4.6. Effects of Heat, pH, and Metal Ions on Antibacterial Activity

The double agar diffusion method was used for the antibacterial experiment.

CFCS was heated in water baths at 65 °C, 85 °C and 100 °C for 10 min and 30 min, respectively. The unheated CFCS (initial CFCS) was used as the blank control to determine the effect of temperature on the antibacterial activity of CFCS.

The initial pH value of CFCS was recorded. NaOH and HCl were used to adjust the pH value of CFCS to 2, 3, 4, 5, 6, 7 and 8. Lactic acid was used to adjust the pH value of MRS broth medium to 2, 3, 4, 5, 6, 7 and 8 as the control. CFCS (initial CFCS) without adjusting the pH value was used as the blank control to determine the effect of the pH value on the antibacterial active substances in CFCS.

NaCl, KCl, CaCl_2_, MgCl_2_, FeCl_3_, FeSO_4_, CuCl_2_, MnCl_2_, AlCl_3_ and ZnSO_4_ were dissolved in CFCS, and the concentrations of each metal ion were 0.02 mol/L, 0.03 mol/L and 0.05 mol/L, respectively. CFCS without metal ions (initial CFCS) was used as the blank control to determine the effect of metal ions on the antibacterial active substances in CFCS.

### 4.7. Mode of Action of L-36 CFCS 

#### 4.7.1. SEM and TEM

SEM treatment: L-36 CFCS was added to the *S. aureus* suspension (10^8^ CFU/mL) to a final concentration of 1 × MIC, and cultured at 37 °C for 12 h. The *S. aureus* without L-36 CFCS treatment was used as control. After washed in PBS buffer solution (0.1 mol/L, pH: 7.4), the cells were fixed 24 h with 2.5% glutaradehyde. Subsequently, the cells were dehydrated 15–20 min by using 30%, 50%, 70%, 85%, 95% and 100% of ethanol. Then, the cells were dried in CO_2_ critical point dryers for 2.5 h, and sprayed with gold powder. Finally, the cells were observed by a variable vacuum ultra-high resolution field emission scanning electron microscope.

TEM treatment: The pretreatment of *S. aureus* of the TEM was the same as that of the SEM. The cells were fixed with 2.5% glutaradehyde, and the samples were sent to Huazhong Agricultural University for processing and photographing.

#### 4.7.2. Effects on Cell Membranes of *S. aureus* by L-36 CFCS

Cell membrane permeability: The effect of L-36 CFCS on cell membrane permeability of *S. aureus* was detected using a fluorescent probe PI [25]. Briefly, *S. aureus* was grown in TSB overnight at 37 °C, and adjusted the concentration to 10^8^ CFU/mL. L-36 CFCS (with a final concentration of 1, 2 and 4 × MIC) was mixed with *S. aureus* suspensions and incubated at 37 °C for 1 h. The *S. aureus* without L-36 CFCS treatment was used as a control. After being washed and resuspended in PBS buffer solution (0.1 mol/L, pH: 7.4), the cells were incubated with PI (final concentration: 5 μg/mL) at 4 °C for 30 min in the dark. Using fluorescence microscopy and automatic microplate reader to observe and detect fluorescence intensity, EX = 535 nm, EM = 615 nm.

Cell membrane potential: The cell membrane depolarization effect of L-36 CFCS on *S. aureus* was determined by DiSC_3_(5) fluorescent probe [26]. *S. aureus* was incubated with different concentrations of L-36 CFCS for 1 h at 37 °C. Then, the *S. aureus* cells were collected and suspended in PBS buffer, and treated with 50 μg/mL of DiSC_3_(5) and incubated in the dark for 1 h at 4 °C. Using fluorescence microscopy and automatic microplate reader to observe and detect fluorescence intensity, EX = 622 nm; EM = 670 nm.

Estimation of pore size in the damaged cell membrane: The size of the pore formed by L-36 CFCS in the cell membrane of *S. aureus* was estimated by detecting the fluorescence intensity of FD4 and FD10 [26]. *S. aureus* was incubated with 1 × MIC concentrations of L-36 CFCS for 1 h at 37 °C. The cells were collected and resuspended with PBS buffer, then FD4 and FD10 (0.1 mg/mL) were added to the cell suspension, and incubated at 37 °C for 1 h. Using the fluorescence microscopy and automatic microplate reader to observe and detect fluorescence intensity, EX = 495 nm, EM = 520 nm.

OD_260_ and OD_280_: The logarithmic *S. aureus* suspension was centrifuged and resuspended in sterile PBS buffer, and the final concentration was 10^8^ CFU/mL. The *S. aureus* suspension was mixed with 10 mmol/L glucose for activation in a 37 °C water bath for 10 min, then L-36 CFCS was added (the final concentration was 1, 2 and 4 × MIC) and cultured at 37 °C. The supernatant was taken at 0 h, 4 h, 8 h, 12 h and 24 h, then measured for the OD_260_ and OD_280_ by automatic microplate reader.

#### 4.7.3. Effects on Subcellular Structure of *S. aureus* by L-36 CFCS

DNA damage: The DNA damage caused by L-36 CFCS in *S. aureus* cells by using DAPI staining [27]. The *S. aureus* cell suspension was treated with different concentrations of L-36 CFCS for 1 h at 37 °C. The cells were harvested, washed, and resuspended in the PBS buffer, then the DAPI was added (final concentration was 1 μg/mL) and incubated in the dark for 20 min at 37 °C. Using the fluorescence microscopy and automatic microplate reader to observe the fluorescence intensity, EX = 358 nm; EM = 641 nm.

ROS production: Fluorescent dye DCFH-DA was used to detect the changes in ROS production in *S. aureus* after treatment with L-36 CFCS [28]. *S. aureus* was incubated with different concentrations of L-36 CFCS for 30 min at 37 °C. Then, the cells were harvested, washed, and resuspended in PBS buffer, treated with 2.5 μg/mL of DCFH-DA and incubated for 20 min at 37 °C. Using a fluorescence microscopy and automatic microplate reader to observe the fluorescence intensity, EX = 480 nm; EM = 525 nm.

### 4.8. Statistical Analysis

All experiments were performed at least in triplicates and all data were analyzed by Graph Pad Prism 5; the data obtained from experiments were presented as mean values and the difference between the control and tested groups was analyzed using the Student’s *t*-test. Significant differences were *p* < 0.05.

## 5. Conclusions

In conclusion, our study suggests that the main antibacterial active substances of L-36 CFCS are peptides. L-36 CFCS are thermostable, active under acidic conditions, insensitive to metal ions and exhibit antibacterial effects by damaging cell membranes and DNA and increasing ROS. Using lactic acid bacteria to inhibit *S. aureus* provides a theoretical basis for the discovery of new antibacterial substances, and will have great significance in the development of antibiotic substitutes, reducing bacterial resistance and ensuring animal food safety.

## Figures and Tables

**Figure 1 molecules-27-07627-f001:**
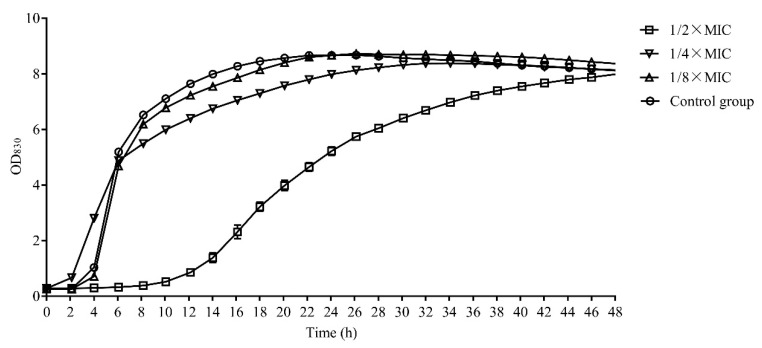
The dynamic growth curve of *S. aureus*.

**Figure 2 molecules-27-07627-f002:**
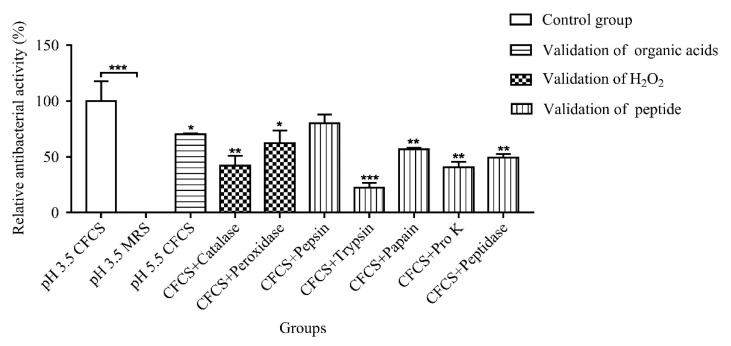
Analysis of antibacterial active substances in CFCS. Statistically significant differences (determined by Student’s *t* test) are indicated as *** *p* < 0.001, ** *p* < 0.01, and * *p* < 0.05 vs. the control.

**Figure 3 molecules-27-07627-f003:**
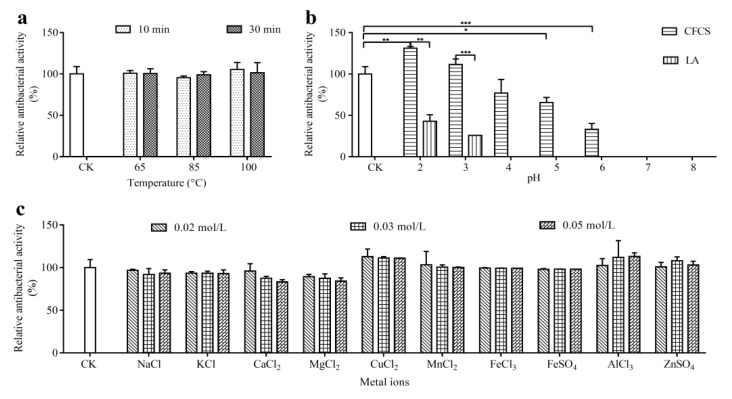
The effects of temperature (**a**), pH (**b**) and metal ions (**c**) on the antibacterial activity of L-36 CFCS against *S. aureus*. CK is untreated L-36 CFCS. Statistically significant differences (determined by Student’s *t* test) are indicated as *** *p* < 0.001, ** *p* < 0.01, and * *p* < 0.05 vs. the control.

**Figure 4 molecules-27-07627-f004:**
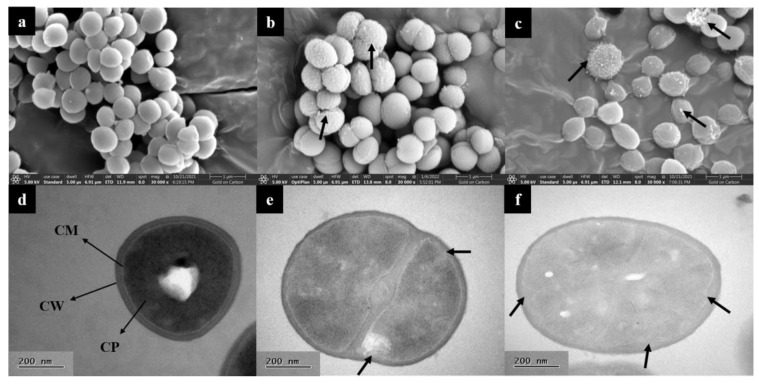
Electron microscopy analysis of *S. aureus*. SEM images of *S. aureus* treated with L-36 CFCS ((**a**) CK; (**b**,**c**) 1 × MIC). Stubby black arrows: intracellular substances and damaged cell. TEM images of *S. aureus* treated with L-36 CFCS ((**d**) CK; (**e**,**f**) 1 × MIC). CW: cell wall; CM: cell membrane; CP: cytoplasm. Stubby black arrows: membrane invagination and damaged.

**Figure 5 molecules-27-07627-f005:**
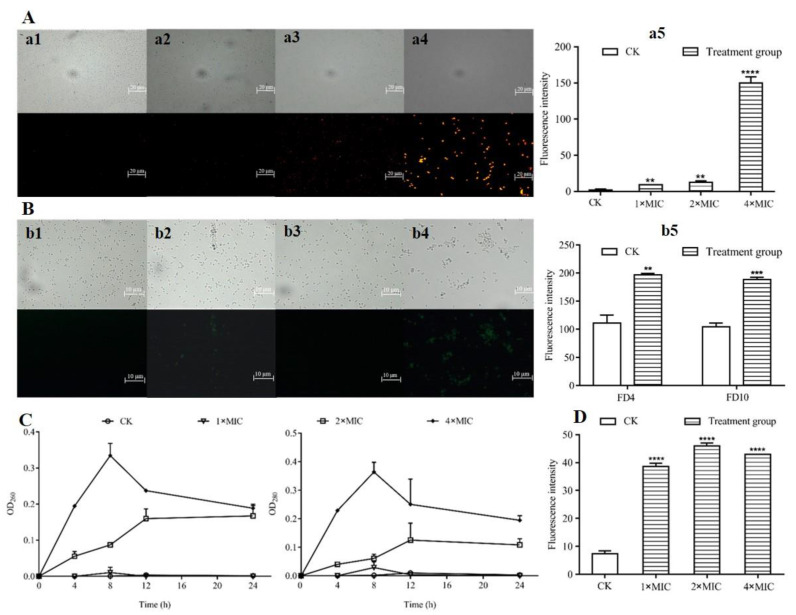
Effects of L-36 CFCS on cell membrane of *S. aureus*. (**A**) PI uptake of *S. aureus*. observed under fluorescence microscopy and fluorescence intensity were determined by automatic microplate reader. (**a1**) Control, (**a2**) L-36 CFCS at 1 × MIC, (**a3**) L-36 CFCS at 2 × MIC, (**a4**) L-36 CFCS at 4 × MIC, (**a5**) is the measurement result of fluorescence intensity corresponding to the experiment, and CK is untreated L-36 CFCS. (**B**) FD4 and FD10 uptake of *S. aureus* observed under fluorescence microscopy and fluorescence intensity were determined by automatic microplate reader. FD4: about 4.0 kDa, 1.4 nm radius; FD10: about 10.1 kDa, 2.3 nm radius. (**b1**) FD4 Control, (**b2**) L-36 CFCS at 1 × MIC, (**b3**) FD10 Control, (**b4**) L-36 CFCS at 1 × MIC, (**b5**) is the measurement result of fluorescence intensity corresponding to the experiment, and CK is untreated L-36 CFCS. (**C**) Effects of L-36 CFCS on leakage of nucleic acids and proteins in *S. aureus* using automatic microplate reader. CK is untreated L-36 CFCS. (**D**) Effects of L-36 CFCS on transmembrane electrical potential of *S. aureus* analyzed using automatic microplate reader. CK is untreated L-36 CFCS. Statistically significant differences (determined by Student’s *t* test) are indicated as **** *p* < 0.00001, *** *p* < 0.001, and ** *p* < 0.01 vs. the control.

**Figure 6 molecules-27-07627-f006:**
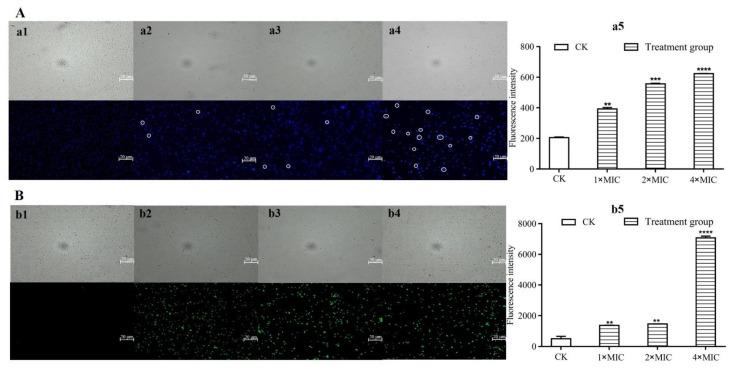
Effects of L-36 CFCS on subcellular structure of *S. aureus*. (**A**) DAPI uptake of *S. aureus* observed under fluorescence microscopy and fluorescence intensity were determined by automatic microplate reader. (**a1**) Control, (**a2**) L-36 CFCS at 1 × MIC, (**a3**) L-36 CFCS at 2 × MIC, (**a4**) L-36 CFCS at 4 × MIC, (**a5**) is the measurement result of fluorescence intensity corresponding to the experiment, and CK is untreated L-36 CFCS. (**B**) DCFH-DA uptake of *S. aureus* observed under fluorescence microscopy and fluorescence intensity were determined by automatic microplate reader. (**b1**) Control, (**b2**) L-36 CFCS at 1 × MIC, (**b3**) L-36 CFCS at 2 × MIC, (**b4**) L-36 CFCS at 4 × MIC, (**b5**) is the measurement result of fluorescence intensity corresponding to the experiment, and CK is untreated L-36 CFCS. Statistically significant differences (determined by Student’s *t* test) are indicated as **** *p* < 0.00001, *** *p* < 0.001, and ** *p* < 0.01 vs. the control.

**Table 1 molecules-27-07627-t001:** Antibacterial effect of L-36 CFCS on clinical isolates of *S. aureus* and MIC.

	Strain Number	Inhibitory Zone Diameter/mm	MIC/mg·mL^−1^
	17-1	22.29 ± 0.15	-
	17-2	22.38 ± 0.69	-
	13-2	19.72 ± 0.69	-
	3-1	20.78 ± 0.62	-
	1-1	20.69 ± 0.10	-
Clinical isolates	16-1	18.35 ± 0.72	-
	3-2	18.74 ± 0.42	-
	9-2	21.73 ± 0.27	-
	13-1	17.99 ± 0.45	-
	16-2	17.67 ± 0.28	-
Standard strain	ATCC 29213	20.40 ± 0.56	31.25

“-”: No test.

## Data Availability

Not applicable.

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
