# Peer review of "Antibacterial Effect of Cell-Free Supernatant from Lactobacillus pentosus L-36 against Staphylococcus aureus from Bovine Mastitis"

_molecules, 2022, doi:10.3390/molecules27217627_

Round 1
Reviewer 1 Report
The manuscript describes the use of Lactobacillus cell-free supernatant against S. Aureus in the treatment of bovine mattitis.
Mastitis is a common disease that impacts animal health and the economy. Its development causes losses in the dairy and derivatives industry. Its fight with antibiotics is expensive, and over time it causes drug resistance. Therefore, finding an alternative treatment is a research goal, and here the possibility of using milk supernatant as an antibacterial agent is presented.
The proposal is very interesting and involves the reuse of tailings from the dairy industry. The results, within my analysis, are well discussed and the idea can thrive.
Could the authors explain the idea of placing metal ions in the supernatant medium? And what are the criteria for selecting precursor salts of metal ions?
Author Response
To Reviewer #1:
Mastitis is a common disease that impacts animal health and the economy. Its development causes losses in the dairy and derivatives industry. Its fight with antibiotics is expensive, and over time it causes drug resistance. Therefore, finding an alternative treatment is a research goal, and here the possibility of using milk supernatant as an antibacterial agent is presented.
The proposal is very interesting and involves the reuse of tailings from the dairy industry. The results, within my analysis, are well discussed and the idea can thrive.
- Could the authors explain the idea of placing metal ions in the supernatant medium?
Responses: Thanks. We found that metal ions have a certain effect on the antibacterial substances of CFCS in practical application through literature survey. In order to make CFCS can be widely used in the future, the influence of metal ions on its antibacterial activity was explored.
2) And what are the criteria for selecting precursor salts of metal ions?
Responses: Thanks. We choose metal ions that other researchers have used, some of which are likely to be encountered in practical applications, and some of which are present in cells to support normal life activities.
Reviewer 2 Report
The manuscript aims in the characterization of anti-staphylococcal effect of cell‑free supernatant from Lactobacillus pentosus L-36. Although the manuscript describes an interesting scientific issue it was very difficult to read. Some sentences are completely incomprehensible. For example: “Secondly, the physicochemical properties of the antibacterial active substances in L-36 CFCS were studied from heat, pH, and metal ions, respectively. (Line 13-14)“ or “…that no green fluorescence was not observed in S. aureus that were untreated with…(line 136-137)”. There are a lot of inaccuracies in results section which was rather detracting. Additionally, the manuscript should be carefully proofread prior to submission to correct for several stylistic errors.
Main specific comments/remarks:
1. Table 1. - Antibacterial effect of L-36 CFCS was performed by disc diffusion method and by MIC determination. Why are there no MIC results for clinical isolates? Presented MIC unit is mg.mL-1. How is it possible when you used cell-free culture supernatants (CFCS), liquid extracts to analyse antibacterial effect? No control antibacterial compound was used (e.g. ciprofloxacin).
2. Line 65-66: “The growth of S. aureus was completely inhibited when L-36 CFCS with 1×MIC concentration were treated (Figure 1).” The experiment was inadequately planned. The results are obvious because you used concentration equal to MIC. According to your MIC definition (line 262-263) “The MIC represents the bacteriocin concentration at which 100% of growth is inhibited. “ All cells are dead at that concentration and no dynamic growth could be observed. You should use concentrations like 1/8MIC, 1/4MIC, 1/2MIC. Moreover, what strain did you use, standard ATCC?
3. Line 90 “The antibacterial effect of MRS broth medium…”?
4. Figures titles lack more detailed description of all used captions e.g. Figure 3 – what is CK, LA?
5. Incorrectly described Figures e.g. “Figure 2 Analysis of antibacterial active substances in CFCS.” Figure 2 illustrates the effect of different enzymes on antibacterial activity of CFCS.
6. Line 155: “Effects of CFCS on leakage of nucleic acid proteins in S. aureus” What does it mean? Proteins or nucleic acids?
7. The leakage of nucleic acids and proteins analysis revealed that at time 0h the observed level of OD260nm and OD280nm was very high. Is it possible?
8. The effect of CFCS on membrane potential of S. aureus was analysed at very high concentrations (1MIC, 2MIC, 4MIC) at which significant disruption of cell membrane is expected. The effect was not concentration dependent. So it is unclear whether the destruction of the membrane changed the potential or the effect of the compound was observed.
9. DNA damage analysis – DAPI shouldn’t be used for that analysis. It is a blue-fluorescent DNA stain and it does not differentiate between the damaged and “normal” DNA.
10. Figure 6. - the descriptions for (A) and for (B) were mixed up. For DAPI staining no differences between blue and blue-green fluorescence (as called by the authors) can be seen. No description for e)
11. Line 201 “The antibacterial components of L-36 CFCS were peptides, H2O2, and organic acids, of which peptides were the most important.” On line 70-71 authors stated: ” …indicating that the main antibacterial active substances were not organic acids.” So are there organic acids or not?
Author Response
To Reviewer #2:
1) Table 1. - Antibacterial effect of L-36 CFCS was performed by disc diffusion method and by MIC determination. Why are there no MIC results for clinical isolates? Presented MIC unit is mg.mL-1. How is it possible when you used cell-free culture supernatants (CFCS), liquid extracts to analyse antibacterial effect? No control antibacterial compound was used (e.g. ciprofloxacin).
Responses: Thanks. Due to the differences among clinical isolates and lack of representation, only MIC values of ATCC 29213 were measured. The unit of MIC refers to the writing method of other articles. Because L-36 with the best antibacterial effect was obtained through mass screening in the preliminary work, antibacterial compound were not used as control.
2) Line 65-66: “The growth of S. aureus was completely inhibited when L-36 CFCS with 1×MIC concentration were treated (Figure 1).” The experiment was inadequately planned. The results are obvious because you used concentration equal to MIC. According to your MIC definition (line 262-263) “The MIC represents the bacteriocin concentration at which 100% of growth is inhibited. “ All cells are dead at that concentration and no dynamic growth could be observed. You should use concentrations like 1/8MIC, 1/4MIC, 1/2MIC. Moreover, what strain did you use, standard ATCC?
Responses: Thank you for your suggestion. We have tested the effect of 1/8MIC, 1/4MIC and 1/2MIC concentrations on S. aureus, and the Figure 1 has revised. We used S. aureus ATCC 29213. 3) Line 90 “The antibacterial effect of MRS broth medium…”?
3) Line 90 “The antibacterial effect of MRS broth medium…”?
Responses: I apologize for not being clear. Line 90 “The antibacterial effect of MRS broth medium…” have revised as “The antibacterial effect of pH 2 and 3 MRS broth medium (adjust with lactic acid)”
4) Figures titles lack more detailed description of all used captions e.g. Figure 3 – what is CK, LA?
Responses: I am sorry for my carelessness. CK is untreated L-36 CFCS, it has been added in the figure note.
5) Incorrectly described Figures e.g. “Figure 2 Analysis of antibacterial active substances in CFCS.” Figure 2 illustrates the effect of different enzymes on antibacterial activity of CFCS.
Responses: I'm sorry the picture doesn't show the result visually. Figure 2 is to explore the antibacterial active substances in CFCS. It is not only treated with enzymes, but also determined whether organic acid is an antibacterial active substance by adjusting pH value. Figure 2 has revised in the article.
6) Line 155: “Effects of CFCS on leakage of nucleic acid proteins in S. aureus” What does it mean? Proteins or nucleic acids?
Responses: Thanks. Detection of nucleic acid and protein leakage. This title have revised as following: “Effects of CFCS on leakage of nucleic acid and proteins in S. aureus”.
7) The leakage of nucleic acids and proteins analysis revealed that at time 0h the observed level of OD260nm and OD280nm was very high. Is it possible?
Responses: Thank you for your suggestion. It is possible that the substances in CFCS affect the test results, and we should remove the interference, and the Figure 5 has revised.
8) The effect of CFCS on membrane potential of S. aureus was analysed at very high concentrations (1MIC, 2MIC, 4MIC) at which significant disruption of cell membrane is expected. The effect was not concentration dependent. So it is unclear whether the destruction of the membrane changed the potential or the effect of the compound was observed.
Responses: Thank you for your suggestion. We studied the change of membrane potential in order to verify the conclusion that the antibacterial active substances in CFCS act on the cell membrane in many aspects.
9) DNA damage analysis – DAPI shouldn’t be used for that analysis. It is a blue-fluorescent DNA stain and it does not differentiate between the damaged and “normal” DNA.
Responses: Thanks. According to the literature, DNA fragmentation occurs in the late phase of apoptosis and when DAPI dye is combined with it, it emits blue-green or bright blue fluorescence, and the fluorescence intensity increases. In this experiment, this feature of DAPI was used to explore whether DNA was damaged.
- Wu X Z, Chang W Q, Cheng A X, et al. Plagiochin E, an antifungal active macrocyclic bis (bibenzyl), induced apoptosis in Candida albicans through a metacaspase-dependent apoptotic pathway[J]. BBA)-Gen Subjects, 2010, 1800(4): 439-447.
- Zeng H, Li T, Tian J, et al. TUBP1 protein lead to mitochondria-mediated apoptotic cell death in Verticillium dahliae[J]. Int J Biochem Cell Biol, 2018, 103: 35-44.
10) Figure 6. - the descriptions for (A) and for (B) were mixed up. For DAPI staining no differences between blue and blue-green fluorescence (as called by the authors) can be seen. No description for e)
Responses: I am sorry for my carelessness. Descriptions for (A) and for (B) have been corrected. Because the cells of S. aureus were too small, the difference in fluorescence could not be clearly seen, but the fluorescence of cells treated with 4MIC was significantly brighter. At the same time, the corresponding fluorescence intensity measurement results also verified this phenomenon. e) is the measurement result of fluorescence intensity corresponding to the experiment, which has been added in the figure note.
11) Line 201 “The antibacterial components of L-36 CFCS were peptides, H2O2, and organic acids, of which peptides were the most important.” On line 70-71 authors stated: ” …indicating that the main antibacterial active substances were not organic acids.” So are there organic acids or not?
Responses: Thank you for your suggestion. Line 201 “The antibacterial components of L-36 CFCS were peptides, H2O2, and organic acids, of which peptides were the most important.” have revised as “LAB exert antibacterial effects by producing organic acids, H2O2, and bacteriocins. By verifying one by one, it was found that the antibacterial active substances of L-36 were peptides and H2O2, among which peptides were the main ones.”
Round 2
Reviewer 2 Report
Thank you very much for the response to the reviewers comments. I believe that the quality of the manuscript has been improved.